# Cumulative SARS-CoV-2 mutations and corresponding changes in immunity in an immunocompromised patient indicate viral evolution within the host

Sissy Therese Sonnleitner [1,2,6 ✉], Martina Prelog[3,6], Stefanie Sonnleitner[1], Eva Hinterbichler[1], Hannah Halbfurter[1], Dominik B. C. Kopecky[1], Giovanni Almanzar[3], Stephan Koblmüller [4], Christian Sturmbauer[4], Leonard Feist[5], Ralf Horres[5], Wilfried Posch [2] & Gernot Walder[1]

Different scenarios explaining the emergence of novel variants of concern (VOC) of the severe acute respiratory syndrome coronavirus 2 (SARS-CoV-2) have been reported, including their evolution in scarcely monitored populations, in animals as alternative hosts, or in immunocompromised individuals. Here we report SARS-CoV-2 immune escape mutations over a period of seven months in an immunocompromised patient with prolonged viral shedding. Signs of infection, viral shedding and mutation events are periodically analyzed using RT-PCR and next-generation sequencing based on naso-pharyngeal swabs, with the results complemented by immunological diagnostics to determine humoral and T cell immune responses. Throughout the infection course, 17 non-synonymous intra-host mutations are noted, with 15 (88.2%) having been previously described as prominent immune escape mutations (S:E484K, S:D950N, S:P681H, S:N501Y, S:del(9), N:S235F and S:H655Y) in VOCs. The high frequency of these non-synonymous mutations is consistent with multiple events of convergent evolution. Thus, our results suggest that specific mutations in the SARS-CoV-2 genome may represent positions with a fitness advantage, and may serve as targets in future vaccine and therapeutics development for COVID-19.

[1] Infektiologie Tirol, Department of Virology, 9931, Unterwalden 30, Außervillgraten, Austria. [2] Institute of Hygiene and Medical Microbiology, Medical University of Innsbruck, 6020 Innsbruck, Austria. [3] Pediatric Rheumatology/Special Immunology, Department of Pediatrics, University Hospital Wuerzburg, Josef-Schneider-Str. 2, Wuerzburg, Germany. [4] Institute of Biology, University of Graz, Universitätsplatz 2, 8010 Graz, Austria. [5] GenXPro GmbH, Altenhoeferallee 3, 60438 Frankfurt am Main, Germany. [6] These authors contributed equally: Sissy Therese Sonnleitner, Martina Prelog. ✉email: sissy.sonnleitner@infektiologie.tirol

In December 2019 the Wuhan Municipal Health Commission (China) reported a cluster of cases of pneumonia of unknown etiology to the WHO China Country Office. By the beginning of 2020 it was confirmed that a novel coronavirus later named severe acute respiratory syndrome coronavirus (SARS-CoV-2), was the causative agent[1]. SARS-CoV-2 spreads easily and effectively among human beings with a basic reproduction number ($R$0) of >2[2,3]. Following this rapid human-to-human transmission and intercontinental spread the WHO declared a global pandemic in March of 2020. The first cases in Austria were reported in Ischgl, Tyrol, as early as February 2020—and East Tyrol was considered one of the first hotspot areas in Central Europe.

While mutations are common in RNA viruses and mostly will not make a significant difference, some mutations proved to provide SARS-CoV-2 with a selective advantage, such as increased transmissibility or increased escape from specific antibodies[4–8]. Those variants with proven or suspected immune escape mutations were deemed variants of concern (VOC) or variants of interest (VOI), respectively, and require close monitoring (https://www.who.int/en/activities/tracking-SARS-CoV-2-variants/). The spread of the first described variant of concern (Alpha variant, B.1.1.7, VOC) was confirmed early in Austria and increased from 0.7% in January to > 99% in April 2021 in the study area of East Tyrol (https://www.ages.at/themen/krankheitserreger/coronavirus/sars-cov-2-varianten-in-oesterreich/). As of December 2020, the European Center for Disease Prevention and Control (ECDC) noticed a sharp decline of the Alpha variant in the European Union (EU; 14.5%) followed by a fast spread of the Delta variant (B.1.617.2) in spring, 2021, plus a very small percentage of the Gamma variant (P1 or B.1.1.28.1, 0.3%) as well as other variants (0.3%). By January 2022, these variants have been largely replaced by the Omicron variant (B.1.1.529; BA.1) in the study area. The Omicron variant now accounts for > 95% of cases. Common in these successful variants are a few specific genomic changes that give them a decisive benefit in terms of transmission or replication, e.g. in the pharynx. The most prominent spike substitutions with immune escape effect can be found in several VOC or VOI. For example, N501Y in the Alpha, Beta and Omicron variant, a nucleotide exchange at position E484 to K in the Alpha- (in a subvariant), Beta and Gamma variants, to Q in a Delta subvariant or to A in both Omicron subvariants BA.1 and BA.2. Furthermore, P681H is found in the Alpha and Omicron variant as well as in the VOI B.1.1.238 and the spike deletions 144/145 described as recurrent deletion regions, since they occurred multiple times[9,10].

The origin of immune escape variants is still a matter of speculation. Several hypotheses take zoonotic origin, selective pressure during treatment with antiviral drugs, monoclonal antibodies or convalescent plasma into consideration and a few studies point to the significance of the exceptional intra-host environment of immunocompromised patients to explain the evolution of immune escape variants[11–13]. The cases involving immunosuppressed patients showed especially long-lasting viable viral shedding of SARS-CoV-2 for a period of more than four months[11–13]. Two of the patients treated with monoclonal and convalescent plasma showed unusually high numbers of nucleotide changes and deletion mutations[12,13], among which was the already described immune escape mutation S:del69/70[13].

Here, we describe the case of an immunocompromised patient with lymphoma who showed persistently high pharyngeal viral loads of SARS-CoV-2 for seven months. In this period of prolonged viral shedding, the strain convergently developed a number of mutations which to a high degree have already been described in the context of variants of concern. The study shows the chronology of the evolution of intra-host mutations, which can be seen as the straight mutational response of the virus to

specific antibodies and should therefore be given special attention in the rating of immune escape mutations of SARS-CoV-2. Our study reveals immunocompromised patients as a potentially new source of virus variants and therefore, emphasises the need to globally give this vulnerable group priority for vaccination.

## Results

**Clinical presentation of an immunocompromised individual persistently infected with SARS-CoV-2.** In August 2015, a female patient in her 60s was diagnosed with stage IVa small cell lymphocytic lymphoma, complicated by a temporary reactivation of Epstein-Barr virus (EBV) with reactive splenomegaly and rapid nodal progression. Beginning in June 2016, she was given six cycles of Rituximab and Bendamustine, which led to remission. In October 2019, the patient suffered a relapse with washout and 90% bone marrow infiltration (B-CLL Binet B or RAI III), accompanied by pronounced symptoms and antibody deficiency. Beginning in May 2020, another round of therapy with Rituximab and Bendamustine was administered. It was completed in November 2020 after six cycles. At that time, the leukocyte count was in the lower normal range at 4200/μL, platelets 136,000/μL, the immunoglobulins were clearly reduced (IgG 249 mg/dL, IgA 3 mg/dL, IgM 12 mg/dL).

Four days after the last chemotherapy—mid November 2020—the patient fell ill with fever, cough, headache and pain, but neither loss of taste nor smell. SARS-CoV-2 was detected in the throat swab by RT-PCR. The patient was in quarantine for 10 days; a final RT-PCR control was not carried out. Due to persistent fatigue, recurrent fever episodes and persistent cough with non-purulent secretion, the patient was again admitted to the hospital in the middle of January 2021 and RT-PCR was again positive for SARS-CoV-2. At the same time there was a recurrence of EBV. The patient was enrolled for an inhalation therapy with N-chlorotaurine (3 times daily inhalation of 10 mL of N-chlorotaurine for 3 min and 10–14 days)[14,15] and received 15 g intravenous immunoglobuline (IVIG, Intratect, Biotest Pharma GmbH, Dreieich, Germany) on admission. Ten days later, with a non-specific antibody therapy against SARS-CoV-2 having been carried out—blood test results were unchanged (leukocytes 5500/μL, platelets 102,000/μL (decreased), IgG 301 mg/dL, IgA 3 mg/dL, IgM 25 mg/dL), but the leukocyte typing showed a 70% decline of B-CLL and a reduction of CD4+ helper T cells. The chest x-ray was normal. The patient suffered from impaired general condition, headache and sore throat and was not able to clear the SARS-CoV-2 infection. IVIG preparations were pooled from donor plasma from pre-pandemic times and did not contain anti-SARS-CoV-1 or anti-SARS-CoV-2 antibodies according to the manufacturer's quality checks (personal communication with the manufacturer). No antiviral therapeutics were administered to the patient at any time of the infection due to the relatively mild course of the infection according to the recommendations of the position paper of the Working Group of Scientific Medical Societies (AWMF) at this time point. A summary of the patient´s medical history is given in Table 1. An increase in leukocytes (21,600/μL) was detected in the final measurement at the end of the infection in June 2021. Finally in June 2021, after six months of persistent viral shedding, two doses of COVID-19-mRNA vaccine (BNT162b2; Comirnaty, BioNTech/Pfizer) were administered. Since the onset of symptoms and the first RT-PCR positive swab the patient was committed to home quarantine in accordance with Austrian law. When symptoms did not clear after a month, home quarantine was slightly lightened, but, testing with naso-pharyngeal swabs and subsequent RT-PCR was continued. Persistent viral shedding was determined via qPCR at 25 time points across a 207-day-long

**Table 1 Summary of the patient's demographic data and latest medical history.**

| | | | |
|---|---|---|---|
| Age | 60–70 | | |
| Gender | Female | | |
| 2015 Aug | Diagnosis of IV a small cell lymphocytic lymphoma | | |
| 2016 June | Temporary reactivation of EBV | | |
| 2016 June | 6 cycles of Rituximab and Bendamustine | | |
| 2019 Oct | relapse | | |
| 2020 May | 6 cycle Rituximab and Bendamustine | | |
| 2020 Nov | End of the last cycle of chemotherapy | Leukocytes: | 4200/µL |
| 2020 Nov | SARS-CoV-2 positive | Platelets: | 136,000/µL |
| | | IgG: | 249 mg/dL |
| | | IgA: | 3 mg/dL |
| | | IgM: | 12 mg/dL |
| 2021 Jan | Reactivation of EBV | Leukocytes: | 5500/µL |
| 2021 Jan | IVIG therapy | platelets: | 102,000/µL |
| | | IgG: | 301 mg/dL |
| | | IgA: | IgA 3 mg/dL |
| | | IgM: | IgM 25 mg/dL |
| 2021 June | BNT162b2; Comirnaty, BioNTech/Pfizer, 2 doses at intervals of 3 weeks | Leukocytes: | 21,600/µL |

period of observation (Fig. 1). After seven months of continuous COVID-19 infection with a mild course of disease and confirmed pharyngeal PCR tests, the viral load became progressively lower and the viral infection was finally cleared. The patient who is now considered to have recovered from SARS-CoV-2 infection gave full written consent for the case to be attended and published.

**Isolation**. Isolation trials were performed from swab samples taken on day 73, 93, 99, 104, 109, 117, 127, 133 and 182 on VeroB4-cells and were successful on days 73, 93, 109 and 127. The isolation success correlated negatively with the Ct-values of the swab ($k = -0.59$).

**Serology**. At the same time points, specific antibodies were investigated using three different serological tools, namely CLIA SARS-CoV-2 TrimericS IgG, microarray immunoblots, neutralization test and Anti-IgG-SARS ELISA with the evaluation of the Anti-IgG-SARS-avidity. An overview of the serological results is given in Table 2. The first serological tests were performed 102 days after symptom onset. At this time, slightly positive values of 17.7 AU/mL were measured in the chemoluminescent immune assay CLIA, as well as borderline neutralization titres of 1:4 in enzyme linked neutralisation assay (ELNA). Humoral immune responses increased in the course of disease and yielded high values of 1320 > 2080 and 1750 AU/mL in CLIA and >1:32 in ELNA, on the days 124, 182 and 205, respectively. Although antibodies increased at day 124 and were maintained, IgG avidity did not mature over time, as the RAI showed no significant increase and stayed in the low range <20%.

The analysis via immunoblot disclosed spike 1 (S1) and the dedicated receptor-binding-domain (RBD) as the main epitopes of the IgG antibodies in the patient's sera, whereas no IgG antibodies could be detected against the region spike 2 (S2) or the nucleocapsid. No specific IgA antibodies were detected.

**SARS-CoV-2 specific T-cell response**. On day 193 no IFN-γ-producing SARS-CoV-2-specific immune cells could be detected in the ELISpot assay (SI = 0.86), although a significant positive reaction against pokeweed mitogen was demonstrated (mean of 213 SFU in the positive control versus mean of 1.4 SFU in the negative control and mean 1.2 SFU cells in the SARS-CoV-2-antigen stimulated wells).

**Humoral immune response did not clear SARS-CoV-2 infection**. The Ct values and numbers of PFU/mL were significantly

lower after day 124. The high titre of IgG antibodies of 1320 AU/mL and a neutralizing antibody titre of 1:32 analyzed by our in-house assay on day 124 was associated with a significant reduction of the viral load but could not clear the infection. We therefore decided to undertake a detailed examination of the specific genetic background of the virus population present including potential intra-host mutational dynamics.

**Mutational intra-host dynamics**. Over the study period of 221 days, 14 haplotypes were sequenced out of naso-pharyngeal samples. The sequences were obtained on day 73, 93, 109, 129, 133, 136, 143, 158, 164, 171, 182, 192 and day 207 of the patient's prolonged infection. The timeline of infection and a chronology of intra-host non-synonymous mutational events are given in Figs. 2 and 3.

The calculation of the pairwise mutation distances did not show higher intra-host evolutionary rates in contrast to overall evolutionary rates of about $8.9 \times 10^{-4}$ substitutions per year[16,17]. The pairwise distance between day 73 and day 171 was $4.4 \times 10^{-4}$ in 98 days, implying a nucleotide substitution rate of $7.5 \times 10^{-4}$.

All NGS sequences were shown to belong to the prevalent Pangolin lineage B.1.1 and the Nextstrain clade 20B.

We became aware of the prolonged viral shedding after about two months and started to regularly sequence the patient's subsequent swabs as of day 73. A listing of all persistent and temporary non-synonymous mutations that the strain has accumulated intra-host and their concordance to VOC and VOI are given in Table 3. Overall, 22 non-synonymous mutations evolved over the study period of 221 days (seven months). Eleven (50%) of these non-synonymous mutations were persistent, whereas eleven (50%) occurred temporarily and were replaced by the wildtype or a different substitution. Seventeen of the 22 non-synonymous mutations evolved in the spike-coding region, eight of those were temporary. Seventeen of the 22 acquired non-synonymous mutations (77.3%) were issued as immune escape mutations by the WHO (https://www.who.int/en/activities/tracking-SARS-CoV-2-variants/) (Fig. 4). Among the persistent non-synonymous mutations in the spike, as many as 88.2% are found in various VOIs or VOCs. All of these changes occurred after the development of high antibody titres. One region continuously showed diffuse mutational changes, with temporary substitutions and deletions, which was ORF1b: position 709–716.

An overview of the intra-host mutational development of SARS-CoV-2 during the study period of 221 days is given in Table 4.

**Chronology of acquired mutations**. In the underlying clinical case the substitutions emerged in the following chronological order:

S:Y144-emerged immediately after the increase of the specific antibody titre at day 117 as a temporary mutation, followed by E484Q (day 129), which could not assert itself against E484K and was displaced at least seven days later (day 136). Furthermore, we found the substitutions S:N354K (day 158, 164, 171 und 182), S:R346I (day 164) and ORF1a:T3284I (day 171), S:D950N (day 171) as well as the prominent S:P681H on day 182 (Fig. 2). Three of the six acquired substitutions (50%) have already been described as typical mutations acquired by diverse VOC.

Thirteen of the seventeen acquired substitutions (76.5%) occurred in the genomic spike-coding region, and one each in the regions coding for ORF1a:T3284, ORF3a:V255X (day 73), ORF8:Y73C (day 73) and N:S235F (day 136) (Fig. 2).

Other mutations appeared temporarily and were subsequently replaced by the wildtype variant. Five hitherto undescribed temporary mutations were observed on the days 73

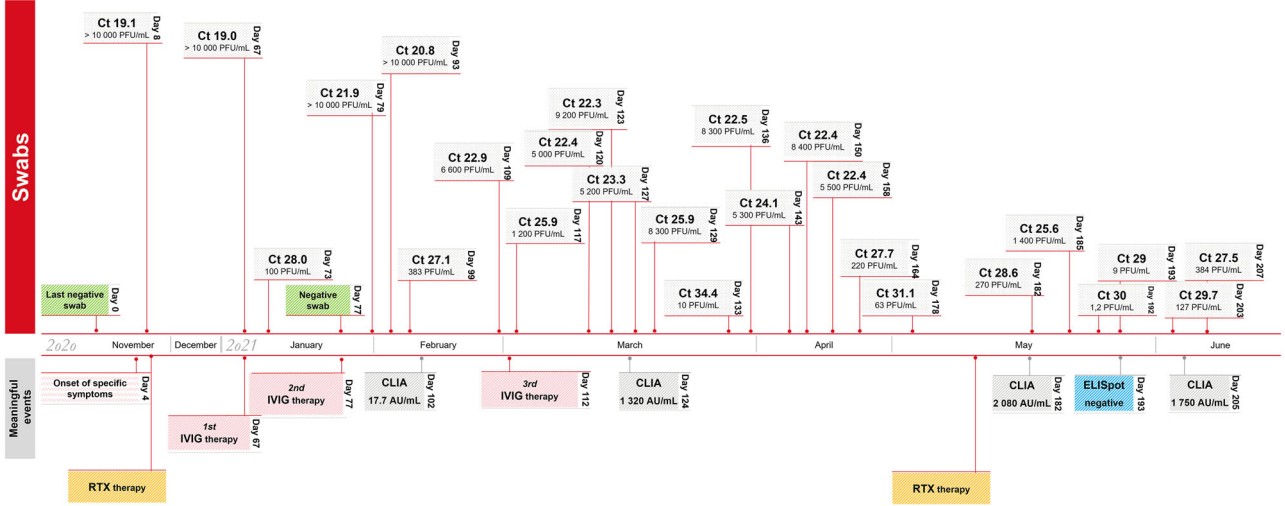

**Fig. 1 Timeline of the prolonged SARS-CoV-2 infection and viral load.** Timeline of the course of disease in an immunocompromised female patient with almost continuous viral shedding throughout the study period of 207 days. Ct, cycle threshold; IVIG therapy, intravenous immunoglobuline therapy; RTX, rituximab and bendamustine therapy. Source data are provided as a Source Data file.

### Table 2 Evaluation of antibody responses at different time points.

| Day | CLIA (AU/mL) | ELNA (titre) | Blot IgG (AU/mL) | Activity (U/mL) | RAI (%) |
|---|---|---|---|---|---|
| 102 | Positive (17.7) | Borderline (1:4) | Negative | 1.28 | n.d. |
| 124 | Positive (1320.0) | Positive (1:32) | Positive S1 (209), RBD (405) | 106.7 | 2.9 |
| 182 | Positive (>2080.0) | Positive (>1:32) | Positive S1 (294), RBD (459) | 145.5 | 17.4 |
| 250 | Positive (1750.0) | Positive (1:32) | Positive S1 (229), RBD (353), S2 (85) | 69.1 | 22.9 |

Overview of the patient's serological data from day 102, 124 and 182. The used diagnostic tools were the LIAISON SARS-CoV-2 TrimericS IgG (DiaSorin S.p.A., Saluggia, Italy) (LIAISON), an Immunoblot called ViraChip assay (Viramed, Munich, Germany), Avidity and an in-house Enzyme-Linked Neutralization Assay (ELNA)[56]. The data obtained from LIAISON SARS-CoV-2 s1/s2 IgG CLIA assay are standardized according to the WHO criteria.
CLIA chemoluminescent immunoassay, AU/mL arbitrary units/mL, ELNA enzyme-linked immunosorbent assay, S1 spike 1, RBD receptor-binding-domain, RAI relative avidity index formula: IgG concentration with chaotropic agent/IgG concentration with PBS.

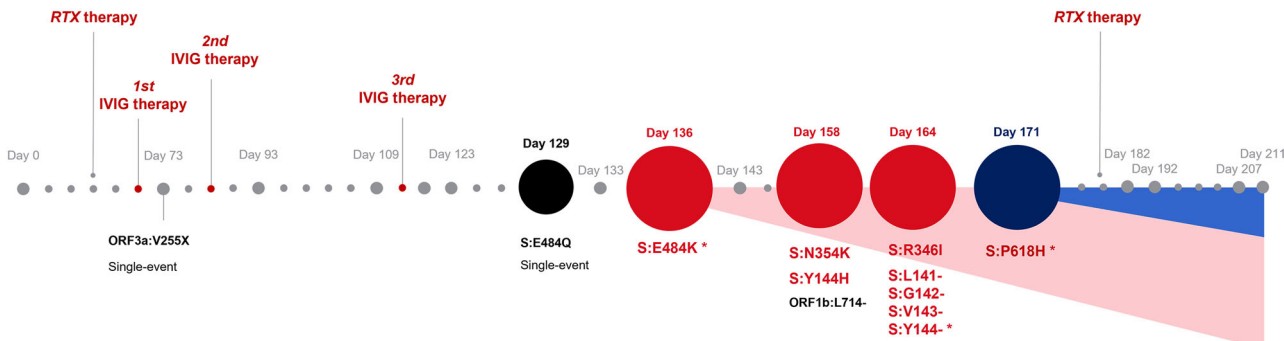

**Fig. 2 Chronology of the evolution of intra-host mutations.** The emergence of intra-host mutations in an immunocompromised patient with adequate humoral and lacking cellular immune response. The study period comprised 140 days of almost permanent viral shedding. High-quality next-generation sequences could be obtained at 14 time-points during the seven-month study period (starting on day 73, ending on day 207 with the last SARS-CoV-2 positive swab) and disclosed the chronological development of mutational events of SARS-CoV-2 as an answer to a unilateral immune response with strong antibody answer but lack of specific T-cells. RTX therapy, rituximab therapy; IVIG therapy, intravenous immune globulin therapy; single-event, temporary mutation. Source data are provided as a Source Data file.

(ORF3a:V255X), 117 (S:A831V), 117–123 (S:Y145X), day 129 (S:Y144H), as well as on day 129 (S:E484Q, S:Y144H) (Fig. 2).

Six temporary mutations have already been described previously, all of them prominent variations known in the context of VOC (https://covariants.org/shared-mutations; Center for Disease Control and Prevention; https://www.cdc.gov/coronavirus/ 2019-ncov/variants/variant-info.html): ORF8:Y73C and S:T716I evolved in the early stage of the infection and are described in the context of the Alpha variant,. S:N501Y, known from the Alpha, Beta, Gamma and Omicron variant, S:del(9) described in the Beta variant, N:S235F, known as a typical substitution of the Alpha variant and S:H655Y, described in the context of the Beta as well

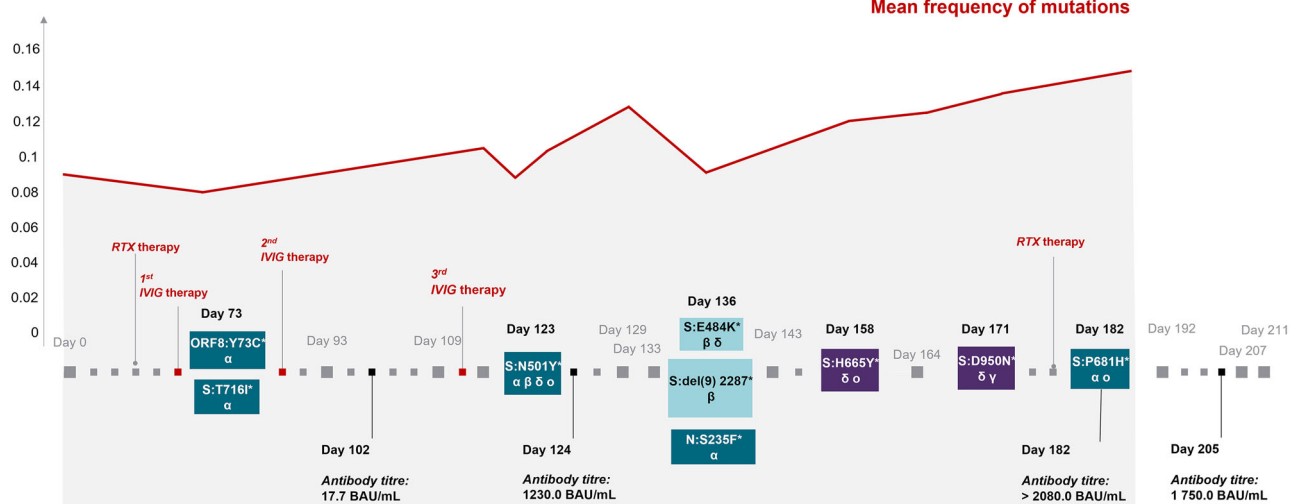

**Fig. 3 Chronology and frequency of the appearance of convergent intra-host mutations.** Evolution of mutations in the region coding for spike in a strain of the clade B.1.1, as they have been proven in identical form in the VOC. α, mutations are described for the Alpha variant B.1.1.7; β, Beta variant B.1.351; δ, Gamma variant B.1.1.28.1; γ, Delta variant B.1.617.2; o, Omicron variant B.1.1.529. The red line graph shows the mean frequency of all mutations at a given day. The development of mutations did not occur linearly, but rather in a fluctuating pattern, with frequent replacement by the wildtype variant. Source data are provided as a Source Data file.

**Table 3 Listing of all persistent and temporary non-synonymous mutations that the strain has accumulated over the 7-month study period.**

|  | n total | VOC/VOI | [%] | n in spike | VOC/VOI | % |
|---|---|---|---|---|---|---|
| Acquired non-synonymous mutations: | 22 | 17 | [77.3] | 17 | 15 | [88.2] |
| Persistent non-synonymous mutations: | 11 | 8 | [72.7] | 9 | 8 | [88.9] |
| Temporary non-synonymous mutations: | 11 | 9 | [81.8] | 8 | 7 | [87.5] |

n total, number of all non-synonymous intra-host acquired mutations; n in spike, number of all non-synonymous intra-host acquired mutations in the spike-coding region; VOC/VOI, number of mutations that are found in comparable expression in a variants of concern (VOC) or a variant of interest (VOI). Source data are provided as a Source Data file.

as Omicron variants. All of these temporarily recurring mutational events did not establish themselves permanently but rather disappeared and/or were dominated again by the wildtype variant. Figure 2 shows the acquired und temporarily acquired mutations of the investigated strain in the spike-coding region and demonstrates the high concordance of the acquired mutations with described VOC, above all the Alpha and the Omicron variant (15/17; 88.2%) and represents the adaptations in the spike-coding region. Further mutations in the genome were ORF1a:T3284I (day 171), ORF8:Y73C (day 73), ORF3a:V255X (day 73), N:S235F (day 136), ORF1b:L714- (day 158). Thirteen of the 17 mutations (76.5%) acquired in the course of the prolonged infectious phase are already described mutations in VOC. Ten of the 17 spike mutations occur in a similar or identical way in the Omicron variant (58.8%). The non-synonymous mutations S:del143, S:del144, S:N501Y, S:H655Y and S:P681H were developed in identical form in the Omicron variant. Additional non-synonymous mutations occurred at the amino acid positions S:142, S:144, S:145, S:484 (twice) in the investigated strain as well as in the Omicron variant, which, however, led to different expressions (S:del142 instead of S:G142D, temporarily both S:Y144H versus S:144del and S:Y145X versus S:145del as well as S:E484Q and S:E484K instead of S:E484A). Overall, 17 of the 22 mutations (77.3%) acquired by the investigated strain convergently evolved in other VOC, mainly in the Alpha and Omicron variants In the spike-coding region, the proportion of acquired mutations identical to mutations of VOC is even higher—15 of the 17 mutations acquired there (88.2%) are found in other VOC. Overall, SARS-CoV-2 developed eleven persistent mutations

during the study period of 140 days as well as eleven temporary mutational events. The chronology of intra-host mutational events is displayed in Figs. 3 and 4. An overview of the total number as well as a characterization of mutations accumulated by the investigated strain during the 7-month study period are shown below.

In the first swab sample, whole genome sequencing did not detect any spike mutations in the investigated strain compared to the reference genome. The first spike variants appeared as E484K on day 133 as a heterozygotic mutation in 41.3% of the targeted reads. On day 136 the proportion of E484K increased to 76% and, after more than seven days (day 143), the new variant dominated with 100%, but decreased to 76.8% again on day 158. On day 171 the spike variant P681H was observed for the first time with a proportion of 24% and dominated within a couple of weeks reaching 100% on day 182. Three of the six acquired substitutions (50%) are previously described substitutions of immune escape variants, namely: S:E484K, S:D950N and S:P681H.

**The fluctuating occurrence of adaptive mutations**. The emergence of adaptive mutations did not occur in a linear fashion, but rather fluctuating. Frequently, new mutations arose at a certain time point to be later replaced by the wildtype variant. As shown in Fig. 4, the mutation rate shows an oscillating course with peaks around day 125, increasing until day 182. Simultaneously, the viral load decreased continuously until the patient had several consecutive negative SARS-CoV-2 RT-PCR tests beginning on day 232 and is therefore considered to be cured from the SARS-CoV-2 infection.

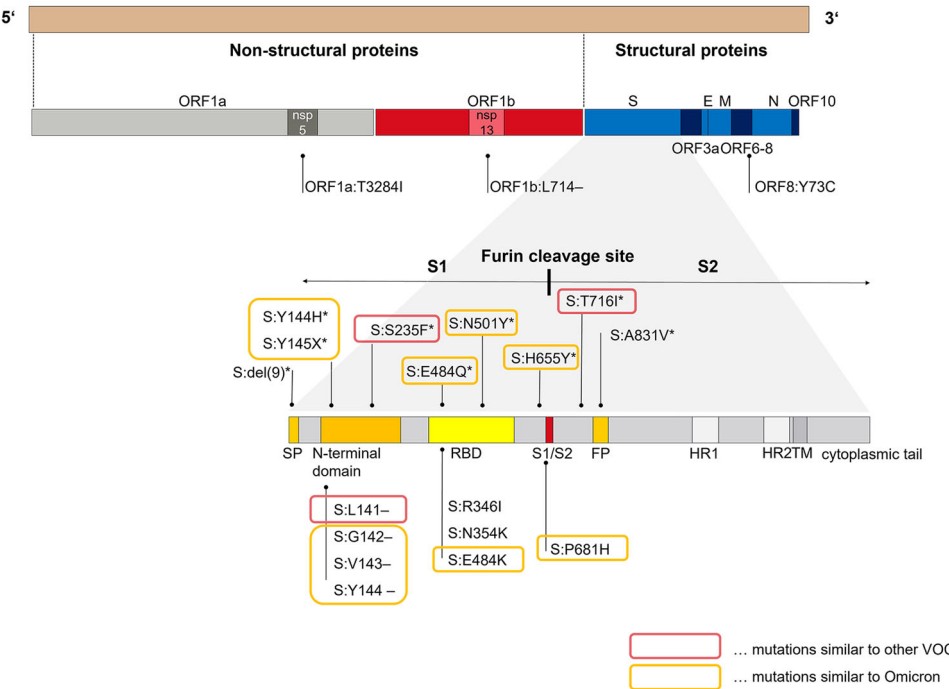

**Fig. 4 Representation of all acquired mutations during prolonged infection.** The acquired and temporarily acquired mutations of the investigated strain in the course of a seven-month-long infection in an immunocompromised person. Overall, 17 persistent or temporary spike mutations were evolved, whereas nine (52.9%) turned out to be temporary and were subsequently replaced by the wild-type variant. *Temporary mutations, S1 spike 1, S2 spike 2, hr heptad repeat, RBD receptor binding domain. The mutations marked in orange are also found in the Omicron variant (B.1.1.529) in similar or identical expression (10 out of 17), mutations marked in red are found in other VOC (3 of 17; 17.6%). All acquired mutations occurred in the regions ORF1a ($n = 1$), ORF1b ($n = 1$), ORF8 (1) and the spike ($n = 17$). Thirteen of the 17 mutations (76.5%) acquired in the course of the prolonged infectious phase are already described mutations in VOC. Source data are provided as a Source Data file.

**Table 4 Overview of (A) substitutions and (B) deletions in the SARS-CoV-2 genome over a 7-month study period in an immunocompromised patient.**

| (A) | Polymorphic substitutions | Strain-specific sub. | Acquired sub. | Temporary mutations acquired and lost sub. |
|---|---|---|---|---|
| | ORF1b:D708A | E:L73F | S:E484K* (day 136) β δ o | ORF8:Y73C* (day 73) α |
| | ORF1b:K709X | N:R203K | S:N354K (day 158) | S:T716I* (day 73) α |
| | ORF1b:Y710X | N:G204R | S:R346I* (day 164) | ORF3a:V255X (day 73) |
| | ORF1b:Y710L | ORF1a:L758V | ORF1a:T3284I (day 171) | S:A831V (day 117) |
| | ORF1b:V711D | ORF1a:P971S | S:D950N* (day 171) δ γ | S:Y145X (day 117–123) o |
| | ORF1b:V711X | ORF1a:M3221I | S:P681H*(day 182) α o | S:N501Y* (day 123) α β δ o |
| | ORF1b:R712X | ORF1a:V3976F | | S:Y144H (day 129) o |
| | ORF1b:V711E | ORF1b:P314L* α δ γ | | S:E484Q (day 129) o |
| | ORF1b:R712X | ORF1b:S598I | | S:del(9) 22287* (day 136) β |
| | ORF1b:L714X | ORF1b:P1000L* δ γ | | N:S235F* (day 136) α |
| | ORF1b:I2568X | S:D614G* α β δ γ | | S:H655Y* (day 158) δ o |
| | | S:A879S | | |
| (B) | Polymorphic deletions | Strain-specific del. | Acquired del. | |
| | ORF1b:K709- | ORF1a:A1204- | ORF1b:L714- (day 158) | |
| | ORF1b:Y710- | ORF1a:E1205- | S:L141- (day 164) o | |
| | ORF1b:V711- | ORF1a:I1206- | S:G142- (day 164) o | |
| | ORF1b:R712- | ORF1a:P1207- | S:V143- (day 164) o | |
| | ORF1b:N713- | ORF1a:K1208- | S:Y144-* (day 171) o | |
| | ORF1b:L714- | ORF1a:E1209- | | |
| | ORF1b:Q715- | ORF1a:E1210- | | |
| | ORF1b:H716- | | | |

Columns 1A and B show genetic variations in ORF1b, a region with repeated changes between substitutions and deletions, i.e. polymorphic substitutions and deletions. Columns 2A and B show substitutions and deletions in comparison to the reference genome Wuhan (GenBank: MN908947.3, RefSeq: NC_045512.2), strain-specific, manifested since the beginning of the infection and maintained throughout the 7-month study period. Columns 3A and B show the chronology of all mutations acquired by SARS-CoV-2 during the intra-host evolutionary process. *Mutations of concern or mutations which are described in the context of immune escape. Column 4 shows temporary mutational events which occurred once and did not occur any more in the following sequence. sub., substitutions; del., deletions α, mutations are described for the Alpha variant B.1.1.7; β, Beta variant B.1.351; δ, Gamma variant B.1.1.28.1; γ, Delta variant B.1.617.2; o, Omicron variants B.1.1.529). Underlined mutation sites are also found in the Omicron variant, but with a different substitution. Source data are provided as a Source Data file.
The table lists all acquired substitutions (Table 4A) and all acquired deletions (Table 4B).

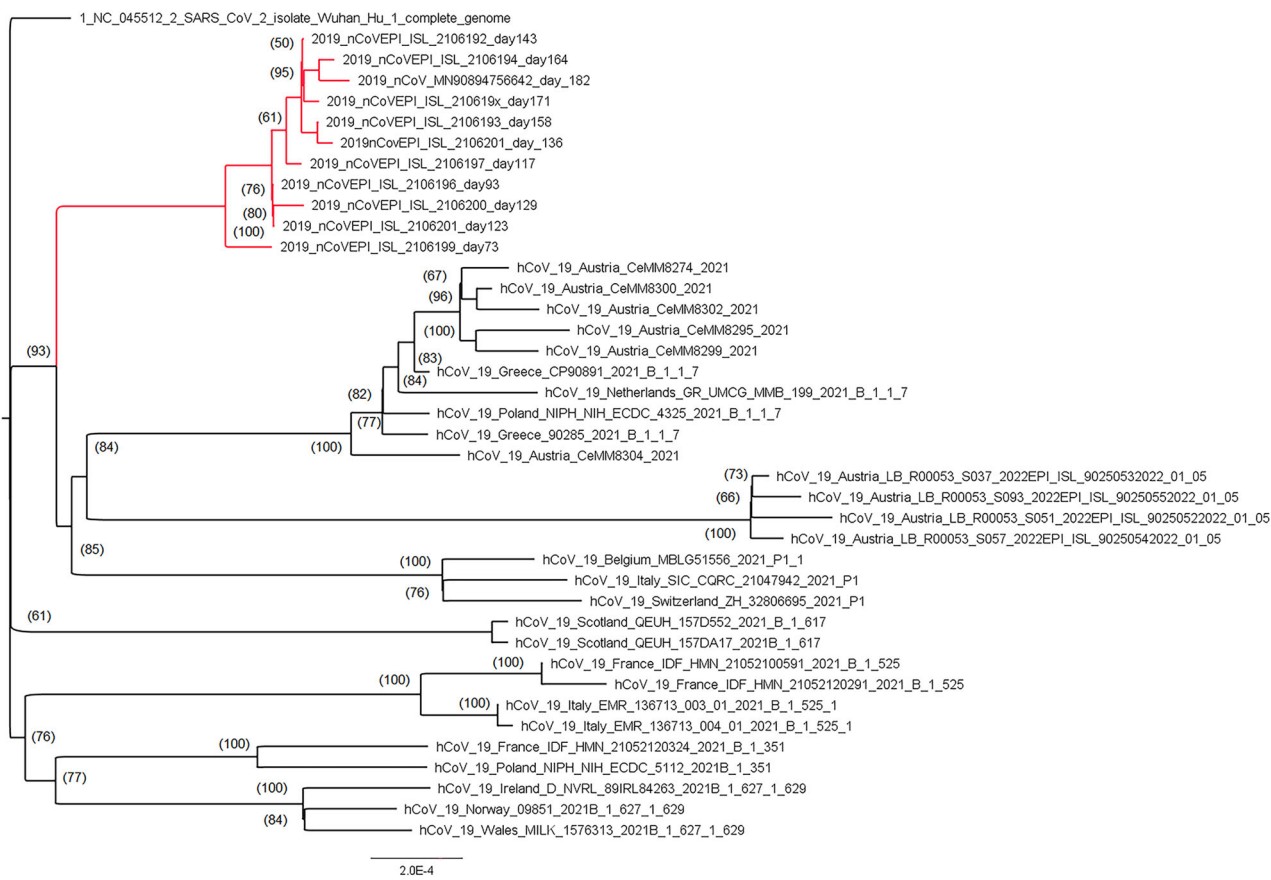

**Fig. 5 Outgroup-routed consensus tree.** The tree is based on 40 SARS-CoV-2 whole genome sequences with 29,806 nucleotide sites. The section highlighted in red shows the monophyletic clade of variants newly formed in an immunocompromised patient, embedded in prominent VOC and typical Austrian strains sequenced in the same investigation period, downloaded from GISAID and supplemented by early Austrian sequences of the Omicron variant in December 2021. Numbers at nodes indicate bootstrap support values (only values >50 are shown).

**Intra-host evolutionary history**. The intra-host evolution of the investigated strain from day 73 and the quasi-species arising from it in the course of the intra-host evolution form a distinct clade in the consensus tree and group together. The clade is placed among others comprised of strains and VOC found in Austria and uploaded to the GISAID platform in the same study period from January to May 2021 (Fig. 5). Early Austrian sequences of the Omicron variant from December 2021 were included subsequently.

## Discussion

In this unique case report we described the dynamics of intra-host mutational events in an immunocompromised patient during a seven-months period of prolonged viral shedding and proven infectivity. We considered the possible influence of a quantitatively strong but regarding binding capacities probably functionally ineffective humoral antibody response and a lack of cellular immune response on the site-directed mutagenesis of SARS-CoV-2.

Our sequencing approach resulted in high-confidence variant identification and robust genome-wide coverage and enabled the establishment of a chronology of immune escape mutations. In addition, previously undescribed site-directed base-exchanges, found in the regions ORF1a and b ($n = 2$; ORF1a:T3284I, ORF1b:L714-), ORF3a (ORF3a:V255X) and spike protein ($n = 2$; S:N354K, S:A831V), were described here. Four different well-established serological methods, namely CLIA SARS-CoV-2 TrimericS IgG, microarray immunoblots, neutralization test and Anti-IgG-SARS ELISA with the evaluation of the Anti-IgG-SARS-avidity gave insights into the humoral immune response and demonstrated the inability to clear the SARS-CoV-2 infection despite positive antibody responses. This may be due to the relatively low neutralizing ability of the detected IgG which is also supported by the low avidity of the specific IgG and impaired avidity maturation over time. Administration of IVIG was not able to enhance the clearance of SARS-CoV-2. Cellular immunity was diminished in this patient and the lack of adapted T cell-mediated immune defence may have contributed to the inefficient clearance. Indications for a reduced SARS-CoV-2 specific cellular immunity were given by the negative IFN-γ ELISpot. The substitution rate for SARS-CoV-2 was estimated as $8.9 \times 10^{-4}$ nucleotides per site per year[16]. This is comparable to previously reported substitution rates of SARS-CoV ($8.0–23.8 \times 10^{-4}$)[18] and MERS-CoV ($11.2 \times 10^{-4}$)[19,20] and comparable to the reported substitution rates for Influenza A ($4–5 \times 10^{-3}$) and Influenza B ($2 \times 10^{-3}$) virus in the haemagglutinine gene[21]. From this substitution rate it can be estimated that SARS-CoV-2 undergoes about one genetic change every other week[16]. In comparison, the nucleotide substitution rate per site and per year for Ebola virus (EBOV Makona) is estimated to be $\sim 1.2 \times 10^{-3}$[22] and for HIV-1 ($3.21–4.06 \times 10^{-3}$)[23]. Interestingly, the evolutionary rate stayed constant throughout the first months of infection and no significant change was measured after the increase of specific antibodies on day 124. We did not find elevated intra-host substitution rates compared to the general rate reported for SARS-CoV-2[16]. This unaltered intra-host evolutionary rate

compared to the global average evolutionary rate suggests that these mutations in an immunocompromised patient, driven by specific antibodies, do not lead to more frequent random genomic changes, but on the contrary to very specific targeted ones.

We assume that the presence of specific antibodies forced directional selection on retaining or regaining infectiousness and thereby strongly favored directional mutations at particular sites, acting as immune escape mutations.

E484K, a substitution in the receptor binding domain (RBD) appeared early in the course of the infection and is described to impair neutralization resistance[24], potentially compromising vaccines effectiveness[4,6,25–32]. E484K is a well-established distinction of the VOC B.1.1.7 with E484K, P.1, P.2, B.1.315, B.1.525, B.1.526 as well as B.1.617.1 (Center for Disease Control and Prevention; https://www.cdc.gov/coronavirus/2019-ncov/variants/variant-info.html). At the same position E484, both subtypes of the Omicron variant have formed the alternative substitution alanine A (Center for Disease Control and Prevention; https://www.cdc.gov/coronavirus/2019-ncov/variants/variant-info.html). On day 158, the Omicron-specific mutation S:H655Y[33,34] could also be detected as a temporary substitution. A further genomic change in the spike-coding region was identified on position S:N354K on day 158 and had never been described before. R346I was detected in the sequence of day 164. This mutation was previously described as a reaction of SARS-CoV-2 after monoclonal antibody treatment, seeming to maintain ACE2 binding activity[35] and has also developed in the VOI Mu, 21H, B.1.621 (https://covariants.org/shared-mutations; Center for Disease Control and Prevention; https://www.cdc.gov/coronavirus/2019-ncov/variants/variant-info.html).

S:D950N arose around day 171 and is as adaptive mutation assigned to the Gamma and Delta variant (Center for Disease Control and Prevention; https://www.cdc.gov/coronavirus/2019-ncov/variants/variant-info.html).

The substitution P681H was observed for the first time in the sequence of day 171, whereby the amino acid histidine (H) first appeared as a polymorphism to become the dominant and finally fixed variant in the course of the next ten days. This transformation from proline (P) to histidine is relatively well studied and implicates a modification in the neighboring furin cleavage site at the junction of the spike protein receptor-binding (S1) and fusion (S2) domains[36].

Another major transformation of the spike protein is the deletion of the amino acids 141 to 144. The deletions Y144/145- on the edge of the spike tip are modifications described in the VOC B.1.1.7 as the recurrent deletion region 2 (rdr2), occurring repeatedly in SARS-CoV-2 variants[9,10].

In our case, the deletions were extended to three more deleted positions on S:141, 142 and 143. S:143del is another analogy to the Omicron variant B.1.1.592.

Nine persistent mutations were found in the spike-coding region. More precisely, four are located in the N-terminal domain (S:L141-, S:G142-, S:V143-, S:Y144-), three in the RBD (S:R346I, S:N354K and S:E484K), both parts of S1 and three are positioned in the region encoding for S2, namely P681H in the immediate neighborhood of the furin cleavage site and S:D905N near heptad repeat 1 as part of the fusion core region.

Of the eleven acquired adaptive mutations, only two were found outside the spike-coding regions, namely L714- in ORF1b (day 158) and T3284I in ORF1a (day 171). ORF1a and ORF1b are coding regions for non-structural proteins (nsp)[37]. ORF1a:T3284I is located in the region encoding for nsp5. Nsp5 is regarded as the main protease, cleaves viral polyprotein and works closely with nsp12 and nsp13. Together, nsp5, 12 and 13 represent the replicase machinery[37–39]. ORF1b:L714- is a deletion in the region coding for nsp13, the enzyme helicase, a main component of membrane-associated replication-transcription complexes[37,40–42].

It is remarkable that nine of the eleven persistent mutations (81.8%) acquired in the course of the prolonged infection had previously been described in the context of immune escape and were assigned to diverse VOC. Our bioinformatic analyses revealed that 75% of the novel mutations in our investigated strain also occur in VOC, whereas the highest concordance was found between the investigated strain and the Omicron variant (50%). Furthermore, we found dynamic mutational events with fluctuations between the wildtype and the variational mutation. Nine of these temporary mutations (9 of 11; 81.8%) have also been described in the context of VOC (Center for Disease Control and Prevention; https://www.cdc.gov/coronavirus/2019-ncov/variants/variant-info.html). In the close proximity of the acquired deletion in ORF1b:L714-, which manifested homozygously, additional conspicuous polymorphic sequences in amino acid position ORF1b:708–716 were found. We measured frequent changes in substitutions and deletions, leaving us with the impression that the constant changes in the genetic structure display immune escape mutations. Also, this hotspot in directional mutations encodes for nsp13, the helicase.

We thus suggest that the accumulated mutations result from an increased selection pressure on the spike, the key to entering the host cell. At the same time a second process takes place intra-host, which exerts increased pressure and enforces continual reconstructions in the nsp13 region. The findings of these temporary mutations, which almost exclusively occurred in the spike region, also fit this pattern very well.

We managed to isolate SARS-CoV-2 from swabs at different time points, which is further evidence for the continuous viability of the virus over the study period, given the evolutionary dynamics of the different sequences. The isolation success correlated negatively with the Ct value; a fact that has already been observed in previous studies[43].

Treatment with Rituximab resulting in depletion of particularly memory and effector B cells by targeting CD20 is known to cause impaired antibody responses[44–46]. As naïve B cell clones are less sensitive to Rituximab treatment due to their lower expression of CD20, a robust immune response can also be assumed for those patients. Immunosuppressive therapy as well as the lymphoma disease itself may have diminished the T cellular axis of immune defence against SARS-CoV-2, which targets infected cells, and loss of control by cytotoxic T cells may have caused the ongoing replication of SARS-CoV-2 in naso-pharyngeal epithelial cells[47,48]. Impaired T cell help may have contributed to the inefficient antibody maturation.

Meanwhile, there are more studies that shed light on the evolution of immune escape variants in immunocompromised patients and support the results of our study[12,49–55]. Nonetheless, our study shows the accumulation of an unusually high number of immune escape mutations in a single patient, which to a strikingly high degree evolved in parallel in various VOC. The chronology of mutation events during seven months of infection shows a rapid accumulation of non-synonymous mutations which in part were persistent, in part temporary or even repeatedly acquired and lost.

In summary, our case report documents the medical phenomenon of persisting SARS-CoV-2 infection in an immunocompromised patient with impaired humoral and cellular immune response. Potential interference of specific antibodies led to a significant reduction in the viral load, but at the same time generated sophisticated escape mechanisms while the cell-mediated immune defence for eradication of the infection was missing. With the aid of NGS, we witnessed the directed mutational changes of SARS-CoV-2, probably facilitated by insufficient humoral immune defence. This led to the formation of highly specific virus variants, highlighting the regions exposed to the

highest intra-host selective pressure. Based on this observation one may hypothesize that immunocompromised patients pose a particular risk to accumulate immune escape mutations and hence be a source for new VOC. This clearly represents an additional risk factor to be considered in the future. Our study also underlines the importance to protect immunocompromised patients from SARS-CoV-2 infection by modified vaccination strategies. Most importantly, the study points out the convergent intra-host evolution of specific mutations in SARS-CoV-2, as they emerged independently in previously described VOCs, VOIs and in the strain we studied. Those specific, convergently evolving mutations reveal those neuralgic positions in the SARS-CoV-2 genome that on the one hand represent its highest fitness advantage, but on the other hand also uncovers its highest vulnerability and should be considered as the probably most important points of attack in future vaccine and therapeutics development.

## Methods

### Immunological diagnostics

*CLIA SARS-CoV-2 TrimericS IgG.* Serological tests were performed using the LIAISON SARS-CoV-2 TrimericS IgG (DiaSorin S.p.A., Saluggia, Italy) (LIAI-SON), an Immunoblot called ViraChip assay (Viramed, Munich, Germany) and an in-house enzyme-linked neutralization assay (ELNA)[56] at day 102, 124, 182 and 205 after the first positive PCR.

The LIAISON SARS-CoV-2 TrimericS IgG is a CLIA (Chemiluminescent Immunoassay) which detects IgG antibodies reactive with the spike protein (S1/S2 domain). The assay was performed on the LIAISON XL Analyzer according to the manufacturer's instructions and gives the arbitrary units per ml (AU/mL) according to the WHO International Standards for the Anti-SARS-CoV-2-immunoglobulin-binding activity (NIBSC 20-136).

**Microarray immunoblots.** The ViraChip assay detects temporal antibody profiles of different immunoglobulin classes against S1, S2, and nucleocapsid (N) as well as against N of the four nonSARS human coronaviruses in a commercial, miniaturized 96 wells protein microarray. The ViraChip assay is a useful tool to identify the epitope-specificity of IgG and IgA in serum samples. The quantitative antibody measurement was performed on a ViraChip Scanner using ViraChip Software.

**Neutralization test.** Neutralization ability of antibodies was determined performing an in-house enzyme-linked neutralization assay (ELNA)[56]. VeroB4 cells (ACC-33, DSMZ) were seeded in flat-bottom 96 well plates (Sarstedt, Germany) with Medium199 (Thermo Scientific Gibco, USA) and 10% fetal calf serum (Thermo Scientific Gibco, USA) at a density of about $10^6$ cells/ml to give a confluent monolayer. Next day, an infectivity titration was carried out to determine 100 tissue culture infectious dose 50% (100 $TCID_{50}$)[57,58]. Sera were heat inactivated by incubation at 56 °C for 30 min. All sera were primarily assessed via a classical plaque-reduction neutralization test (PRNT). To evaluate the cut-off titre for the PRNT, 100 sera of healthy East Tyrolean blood donors from the pre-pandemic years 2012 and 2013 were assessed in SARS-CoV-2 specific PRNT and ELNA. The cut-off titers were set at 1:32 with a viral solution of 100 $TCID_{50}$ for PRNT and 1:4 with a viral solution of $1 \times 10^5$ $TCID_{50}$ for ELNA.

With these evaluated sera, we adapted the PRNT to an ELNA without the need of an apparent cytopathic effect (CPE) and a shorter incubation period of <24 h. For ELNA, sera were titrated in duplicate in twofold dilution steps, starting at a dilution of 1:4 in Medium199 containing 3% fetal calf serum. Equal volumes of virus ($1 \times 10^5$ $TCID_{50}$) and serum dilutions in Medium199 were mixed and subsequently incubated for 1 h at 35 °C in U-bottom 96-well plates (Thermo Scientific Thermo Scientific Nunc, USA). After incubation, a pre-seeded flat-bottom 96-well plate with confluent VeroB4 cells was used, medium was discarded, the incubated mixture of patient's serum and defined virus solution was transferred to each corresponding well of the flat-bottom plate and the plate was incubated for 24 h at 35 °C. Incubation was stopped by discarding supernatant, cells were washed in PBS twice, fixed with ice-cold acetone–methanol (1:1) and frozen for at least 15 min. All steps were performed under strict observation and in compliance with biosafety level 3. The analysis was carried out like an enzyme-linked immunosorbent assay using a BEP III (Siemens, Germany) according to the following steps: blocking (45 min, 37 °C, Candor Biosciences, Germany), washing 3 times (wash pod, Siemens, Germany), anti-SARS-CoV-2 nucleocapsid protein IgG (Bioss bsm-41413M, dilution 1:5000) for 30 min at 37 °C), washing 3 times, adding of horseradish-peroxidase-conjugated goat anti mouse IgG (ABIN376241, dilution 1:5000) for 30 min at 37 °C), washing 3 times, adding substrate tetramethylbenzidine (TMB) and stop solution (Siemens, Germany). The cut-off titer was set by titrating defined negative human sera from volunteers out of healthy Tyrolean blood donors from the year 2009 and was set at 1:4 in combination with the viral dose of $1 \times 10^5$ $TCID_{50}$ and calculated as median optic density minus the

standard deviation. A sample was considered positive when the given optic density was higher than the cut-off titer.

Patients' sera were evaluated in duplicate in ELNA and were scored as follows: titers of 1:4 as weak neutralization; titers of 1:8 or 1:16 as moderate/good and >1:16 as strong neutralizing ability. Sera with single titers between 1:4 and negative were valued as borderline.

**IgG-anti-spike ELISA.** Serum IgG antibodies against SARS-CoV-2 were determined by Serion agile SARS-CoV-2 ELISA with a sensitivity of 96.2% and a specificity of 100% according to manufacturer's instructions (Virion/Serion, Wuerzburg, Germany). Antibody activities above 15 U/mL were considered positive.

**IgG-anti-spike-avidity.** Relative avidity index (RAI) was determined by a modification of the Serion agile SARS-CoV-2 IgG-SARS ELISA using 1 M ammonium thiocyanate ($NH_4SCN$) as a chaotropic agent as described previously[59–61]. RAI values were considered as: RAI > 60% high avidity, 40% <RAI < 60% as moderate, and RAI < 40% as low avidity in reference to other viral infections[62].

**SARS-CoV-2 specific T cell response.** The ELISpot assay was performed using a commercially available precoated human SARS-CoV-2-specific IFN-γ ELISPOT kit according to the manufacturer's protocol (AutoImmun Diagnostika, GmbH, Germany; Cat.no. ELSP 5500). Peripheral blood was collected into tubes coated with lithium-heparin (Vacuette, Greiner bio-one, Austria). PBMCs were separated from plasma and whole blood by gradient density (FicoLite -H, Linaris, Germany). After washing with phosphate-buffered saline (PBS), depleting erythrocytes (RBD-Lyse Buffer Life Technologies, 1xRBC Lysis Buffer 200 mL; Invitrogen eBioscience, USA REF: 00–4333) and washing again with PBS, cells were counted and resuspended in x-vivo medium (X-VIVO TM-10 Serum-free hematopoietic cell medium; BEBP02-055Q, Lonza, Switzerland).

Briefly, a total of $2 \times 10^5$ PBMCs were incubated in duplicate with x-vivo as a negative control, pokeweed mitogen (AutoImmun Diagnostika GmbH, Germany) as a positive control,15–20mer peptide pools for SARS-CoV-2 (AutoImmun Diagnostika GmbH, Germany) and PanCorona (AutoImmun Diagnostika GmbH, Germany) for the four nonSARS human coronaviruses 229E, HKU1, NL63 and OC43 as a control of possible cellular cross-reactive responses. After incubation at 37 °C for 20 h in a sterile and humidified atmosphere, plates were washed with washing buffer (AutoImmun Diagnostika GmbH, Germany) and stained with the kit-specific reagents according to the manufacturer's protocol. Plates were then washed several times under running water and dried overnight. Spot forming units (SFU)/100,000 cells were counted using an automated AID ELISPOT reader system (AutoImmun Diagnostika GmbH, Germany).

The assessment criteria for the ELISpots were a minimum of 50 SFU in the positive control and a maximum of 10 SFU in the negative control according to the manufacturer's definitions[63,64]. When those criteria were fulfilled, the stimulation index (SI) was calculated by dividing the mean SFU numbers in the antigen-specific wells with the mean SFU numbers of the negative control. The test was assessed negative with an SI < 2 according to previous determination of the cut-off by well-defined pre-pandemic PBMC samples and by PBMCs from SARS-CoV-2-naive individuals. The test was suggested to be poorly reactive with an SI between 2 and 7 and reactive with an SI ≥ 7 as defined by the manufacturer[63]. According to standardized laboratory procedures, in each assay, a standard laboratory control sample of a high-reactive and a non-reactive PBMC sample, respectively, was run to determine inter-assay-variations. Only assays with less than two standard deviations of the high-reactive and the non-reactive PBMC control sample, respectively, were defined valid.

**Sample collection.** Nasopharyngeal swabs were taken in a standardized way in home quarantine in the context of primary care by a medical co-worker. All clinical samples and data were collected for routine patient care and for public health interventions. The patient gave full written consent for the case to be attended and published and the study was performed according to the principles of the declaration of Helsinki 2013. Ethical approval to use residual routinely taken serum samples for retrospective analyses was obtained by the Ethics Committee of the University Hospital Wuerzburg (no. 20201105_01).

**RNA extraction.** Nucleic acids were isolated using the MagMAX-96 Total RNA Isolation Kit (Thermo Fisher Scientific, Waltham, Massachusetts, USA; Cat. No. AM1830). Briefly, 200 μL PBS were taken from patient swab sample and mixed with 265 μL binding buffer, 5 μL proteinase K (20 mg/mL) and 5 μL extraction control (Thermo Fisher Scientific, Waltham, Massachusetts, USA; Cat. No. AM1830) according to the KingFisher extraction protocol for 200 μL sample volume (Thermo Fisher Scientific, Waltham, Massachusetts, USA). After incubation at room temperature for at least 15 min, samples were transferred from tubes into 96-well KingFisher deep well plates (Thermo Fisher Scientific, Waltham, MA, USA) containing 280 μL isopropanol and 2 μL Mag-Bind particles per well, using a KingFisher Flex purification system (Cat. No. 5400620).

**RT-PCR**. RT-PCR extracts were evaluated for SARS-CoV-2 by qRT-PCR using the Bio-Rad CFX96 system (Bio-Rad, Germany) with a LightMix Modular Assay kit in accordance with the modified Charité guidelines[65]. 10 μL of extracted RNA were added into 15 μL 4× Reliance One-Step Multiplex Supermix (Bio-Rad, Germany). Each 15 μL mastermix contained 12.5 μL buffer solution, 0.25 μL enzyme mix, 1.75 μL of nuclease-free water and 0.5 μL primer probe wHCoV (E-Gene, as well as N-Gene and Rdrp-Gene for confirmation; TIB Molbiol, Germany, Cat. Nos. 53-0776-96, 53-0775-96 and 53-0777-96). Reactions were incubated at 55 °C for 5 min and 95 °C for 5 min in order to conduct reverse transcription of viral RNA, sample denaturation and enzyme activation. These steps were followed by PCR-amplification including 45 cycles at 95 °C for 5 s, 60 °C for 15 s and 72 °C for 15 s. Cooling was implemented at 40 °C for 30 s. Results were interpreted based on the Second Derivative Maximum (SDM) method. Positive results were confirmed by Rdrp and N-gene[65], samples with an initial Ct value lower than or equal to 37 were assigned to repeated testing including extraction. A Ct value higher than 40 was considered negative. Quantification of the viral load in the swabs was calculated via size standards of 1, 10, 100 and 1000 plaque-forming units (PFU)/mL. Standardization of viral stocks was carried out by virus titration. Isolation was performed on VeroB4 cells as described elsewhere[43].

**Virus titration**. Confluent VeroB4 cells were cultured in Medium199 including 5% FCS in T75 tissue culture flasks (Sarstedt, Germany) and transferred into 96-well tissue culture plates (Sarstedt, Germany). Passage 1 isolates of SARS-CoV-2 were thawed from −80 °C freezer and titrated from 1:10 to 1:10$^{-12}$ in U-shaped 96-well plates (Greiner, Germany) and pipetted into each corresponding well of the 96-well tissue culture plate. Plates were incubated at 36 °C. Three days post infection, incubation was stopped by gently removing the supernatant, washing the cells three times with PBS and fixing cells in 1:1 ice-cold acetone–methanol. For easier optical evaluation, cells were dyed by crystal violet staining and tissue culture infectious dose of 70% (TCID$_{70}$) and PFU were calculated[57].

**Whole genome sequencing and mutational analysis**. Libraries were prepared according to the Ion AmpliSeq SARS-CoV-2 Research Panel (Thermofisher, USA), library construction and sequencing protocol with the Library Kit Plus (Thermo Fisher Scientific, Waltham, Massachusetts, USA; Cat. No. 4488990). The Ampli-cons were cleaned up with AMPure XP beads (Beckman Coulter, Germany) with a 1:1 ratio. The libraries were quantified using the Ion Library TaqMan Quantitation Kit (Cat. No. 4468802), normalizing, pooling and sequencing was performed using an Ion Torrent S5 Plus. Ion Torrent Suite software (v 5.12.2) of the Ion S5 sequencer was used to map the generated reads to a SARS-CoV-2 reference genome (Wuhan-Hu-1; GenBank accession numbers NC_045512 and MN908947.3), using TMAP software included in the Torrent Suite. The following plugins were used: Coverage Analysis (v5.10.0.3), Variant Caller (v.5.12.04) for mutation calls both with "Generic—S5/S5XL (510/520/530)—Somatic—Low Stringency" and "Generic-S5/S5XL (510/520/530)-Germ Line-Low Stringency" default parameters and COVID19AnnotateSnpEff (v.1.0.), a plugin specifically developed for SARS-CoV-2 that can predict the effect of a base substitution. No ultra-deep sequencing was performed and only mutations visible in the stated analysis methods were listed and rated.

FASTA files containing the raw reads were inspected for quality criteria (mapped, targeted, filtered reads, mean depth and uniformity) using Thermofisher Software. Multiple sequence alignments were performed using Unipro UGENE[66] as well as MEGA X[67]. The SARS-CoV-2 genomes were compared to the reference NC 045512.2-Wuhan-Hu-1. Viral genome assembly and screening for distinct mutations was performed online using nextstrain.org (https://github.com/nextstrain/ncov/blob/master/defaults/clades.tsv; https://clades.nextstrain.org/). The identification of pangolin lineages was carried out using Pangolin software, v.2.4.2. (https://pangolin.cog-uk.io/). The generated full-genome sequences are available at Genome Sequence Archive as.bam files under the Submission ID subCRA009527, Accession numbers CRR453213 (day 73), CRR453214 (day 93), CRR453215 (day 117), CRR4536216 (day 123), CRR453217 (day 129), CRR453218 (day 136), CRR453219 (day 143), CRR453220 (day 158), CRR453221 (day 164), CRR453222 (day 171), CRR453223 (day 182).). Additional sequences of frequent Austrian strains and prominent VOC, sequenced in the same study period, were retrieved from the GISAID EpiCoV database[68] to calculate a consensus tree. Indels were coded using 'simple indel coding'[69] as implemented in 2matrix v.1.0[70].

The best-fit model of nucleotide substitutions (TIM2 + F + I) was selected under the Akaike and the Bayesian formation criteria using ModelFinder[71] as implemented in the PhyloSuite Software package[72]. Phylogenomic inference was based on a Maximum Likelihood (ML). An ML tree with 5000 ultrafast bootstrap replicates was inferred in the IQ Tree plugin[73,74] of PhyloSuite.

**Bioinformatics CoV-Seq Workflow**. The frequency of the various mutations and the homology to the most widespread VOC (Alpha-, Beta, Gamma- and Delta-variant) were investigated based on BAM files. Reads from CoV-Seq samples were demultiplexed by using in-house tools. Reads originating from human were filtered out by mapping against hg38 with bwa-mem 0.7.17[75]. All reads not mapping to human were trimmed for adapters und quality by using Cutadapt 3.2[76]. The trimmed reads were mapped with bwa-mem 0.7.17 to the SARS-CoV-2 reference

MN908947.3 from the NCBI. Mutations were called using breseq 0.35.5[77]. Graphics were created using *pandas* 1.2 for Python 3.

**Statistics**. Dichotomous data were evaluated by a chi-squared test or Fisher´s exacta in the case of small group size ($n < 60$) (Microsoft Excel, Microsoft 395 MSO, Windows 10). A two-sided significance level of $p < 0.05$ was used for determining statistical significance. After testing for distribution (Kolmogorov–Smirnov-test), non-parametric continuous independent variables were compared using Mann–Whitney-*U* test for each time point. Dependent non-parametric variables were compared using Wilcoxon-rank test.

**Reporting summary**. Further information on research design is available in the Nature Research Reporting Summary linked to this article.

## Data availability

The generated whole genome sequences of days 73, 93, 117, 123, 129, 148, 164, 171 and 182 are available in the Genome Sequence Archive as.bam files under the bioproject name PRJCA008906 (https://ngdc.cncb.ac.cn/gsa/browse/CRA006527). The sequences are deposited and available under the following accession numbers CRR453213 (day 73), CRR453214 (day 93), CRR453215 (day 117), CRR4536216 (day 123), CRR453217 (day 129), CRR453218 (day 136; https://ngdc.cncb.ac.cn/gsa/browse/CRA009635/CRR463997), CRR453219 (day 143; https://ngdc.cncb.ac.cn/gsa/browse/CRA009635/CRR463998), CRR453220 (day 158), CRR453221 (day 164), CRR453222 (day 171), CRR453223 (day 182). Source data are provided with this paper.

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

## Acknowledgements

We gratefully acknowledge the financial support of the Austrian Research Promotion Agency (FFG), Grant No. 889135. We also thank Tobias Albert for excellent technical assistance, Martin Lamprecht for valuable suggestions and all the kind colleagues from the routine diagnostics team of the Dr. Gernot Walder laboratory and Infektiologie.Tirol for the great support. Special thanks also go to Steven Weiss for proofreading and grammar corrections.

## Author contributions

S.T.S.: Conceptualization, data curation, formal analysis, funding acquisition, Investigation, methodology; writing—original draft preparation. M.P.: Interpretation and discussion of clinical data and laboratory results, writing—original draft preparation, editing; S.S.: Methodology, writing—editing. E.H.: Investigation; methodology; H.H.: Project administration. D.B.C.K.: Visualization, organization. G.A.: Methodology. S.K.: Conceptualization, writing—editing, funding acquisition. C.S.: Conceptualization, writing—editing, funding acquisition. L.F.: Validation, verification. R.H.: Validation, verification. W.P.: Validation, supervision. G.W.: Supervision.

## Competing interests

The authors declare no competing interests.
