## [Peer Review File · Nature Communications]

REVIEWER COMMENTS

Reviewer #1 (COVID-19, convalescent plasma therapy) (Remarks to the Author):

I have read the manuscript named 'First evidence of in-host immune escape mutations of SARS-CoV-2 in an immunocompromised patient: a possible new source of mutations?' with interest.

Below are my comments on the manuscript:

1. Did the authors use antiviral therapy for SARS-CoV-2?
2. The authors has given the antibody titres of the patient in a timeline basis. However, the antibody levels did not reflect the antibody of the patient as the patient was given IVIG three times, on the 67th, 77th and 112nd days. We know that the commercial IVIG products include anti-SARS-CoV-2 antibodies.
3. The authors may add the RTX therapy date as well as IVIG date in the figure that includes the immunoglobulin levels of the patient.
4. The authors suggested in the discussion (line 501-2) that the case report documents a patient with an adequate humoral but impaired cellular immune response. The patient has very low antibody titre (301mg/dl). So, we can not suggest that the patient has adequate humoral immune response. Another point is that the authors did not give any absolute lymphocyte count and lymphocyte subset value. So, they can not indicate accurately that the patient has impaired cellular immune response.
5. Discussion: Line 506-507: The authors have written 'With the aid of NGS, we witnessed the directed mutational changes of SARS-CoV-2 probably enforced by exclusively humoral immune defence without destruction of infected cells by cell mediated immunity.' The authors may change this sentence as 'With the aid of NGS, we witnessed the directed mutational changes of SARS-CoV-2, probably enforced by insufficient humoral immune defence.'

Minor points:

1. Introduction

Line 44: EDCD? Please write before abbreviation (European Center for Disease prevention and control).

Line 57: reconvalescent plasma? What is the difference from convalescent plasma?

Line 61-63: Two of the cases, patients treated....

2. Statistics: $p=0.05$  $p<0.05$,

3. Intratect  Please write IVIG by not indicating the brand.

4. Discussion:

Line 269 (Results), 433, 459, 460: domaine  domain

Line 485: What does nsp stands for?

5. Serology:

Line 264: ELNA, CLIA  Please write before abbreviation.

Reviewer #2 (COVID-19, viral immunity) (Remarks to the Author):

This manuscript describes an interesting case of a single immunocompromised individual who experienced prolonged SARS-CoV-2 infection with shedding of infectious virus. The authors demonstrate time-dependent intra-host evolution of SARS-CoV-2 including the emergence of multiple mutations associated with immune escape that have been found in VOI/VOCs. Additionally, the emergence of escape mutations primarily occurs later in disease in the setting of an improved humoral immune response (based on serology testing) that is sufficient to decrease viral titres without eradication. The main conclusion is that prolonged viral replication in immunocompromised hosts can be a source of viral

evolution, including the selection of specific mutations associated with immune escape.

The mechanisms driving SARS-CoV-2 evolution and the selection of immune escape mutations is of broad interest. However, there are now many reports of SARS-CoV-2 viral evolution in immunosuppressed hosts (For example - PMID 33176080, 33831372, 33915337, 34278371, 34737266, 34431691, 34319130, 33248470, 33545711). Many of these reports have documented the emergence of viral variants harboring mutations associated with VOI/VOCs and/or humoral immune escape, including many mutations described in this manuscript. Given that this phenomenon has now been reported multiple times and many of these reports include detailed immunologic characterizations and have identified the emergence of escape mutations, this case adds support to what is already known rather than describing a new finding.

Other recommendations:

- As there are other reports of in-host evolution of escape mutations during infections in immunosuppressed individuals the title "first evidence" is incorrect**
- SARS-CoV-2 T cell studies should include additional individuals with or without previous SARS-CoV-2 infection as positive and negative controls to ensure that the SARS-CoV-2 peptide pools / assay is functioning as expected.**
- Editing could significantly improve the presentation and clarity of the manuscript**

Point to point response letter concerning the Manuscript (NCOMMS-21-39674A)

Authors: Thank you for considering our manuscript (NCOMMS-21-39674-T), formerly entitled " First evidence of in-host immune escape mutations of SARS-CoV-2 in an immunocompromised patient: a possible new source of mutations?" for publication. The title is now changed to "The mutational steps of SARS-CoV-2 to become like Omicron within seven months: the story of immune escape in an immunocompromised patient" in view of current developments of Omicron and other variants of concern (VOC).

We appreciate the critical comments of the reviewers. We have carefully considered these comments and suggestions and revised the manuscript accordingly. With these improvements, we hope that the current version can meet the Journal's standards for publication. The following is a point-by-point response to all those comments and a list of changes we have made to the manuscript. The new changes and revisions in the manuscript are highlighted in green.

Reviewer comments and point-to-point reply:

Reviewer #1 (COVID-19, convalescent plasma therapy) (Remarks to the Author):

I have read the manuscript named 'First evidence of in-host immune escape mutations of SARS-CoV-2 in an immunocompromised patient: a possible new source of mutations?' with interest. Below are my comments on the manuscript:

1. Did the authors use antiviral therapy for SARS-CoV-2?

Authors: We did not use any antiviral therapy for SARS-CoV-2 and added this information now: "No antiviral therapeutics were administered to the patient at any time of the infection." Page 7, Line 225-6.

2. The authors have given the antibody titres of the patient in a timeline basis. However, the antibody levels did not reflect the antibody of the patient as the patient was given IVIG three times, on the 67th, 77th and 112nd days. We know that the commercial IVIG products include anti-SARS-CoV-2 antibodies.

Authors: Thank you very much for pointing this out. We have taken it into account in the discussion: After identification of the batch used and consultation of the manufacturer, we confirm that pre-pandemic plasma was used for preparation of the IVIG batch administered to our patient. According to the manufacturer's pre-testing of the product, no SARS-CoV-2 specific antibodies were detectable in the batch used at any time. Therefore, we may assume that the antibodies detected by us serologically originate from the patient's own immune defence.

3. The authors may add the RTX therapy date as well as IVIG date in the figure that includes the immunoglobulin levels of the patient.

Authors: Figure 1: We fully agree and included IVIG –and RTX dates in Figure 1. Page 8.

Figure 1: Timeline of the course of disease in an immunocompromised 65-years-old female patient with almost continuous viral shedding throughout the study period of 207 days.

4. The authors suggested in the discussion (line 501-2) that the case report documents a patient with an adequate humoral but impaired cellular immune response. The patient has very low antibody titre (301mg/dl). So, we cannot suggest that the patient has adequate humoral immune response. Another point is that the authors did not give any absolute lymphocyte count and lymphocyte subset value. So, they cannot indicate accurately that the patient has impaired cellular immune response.

*Authors: We fully agree and changed the sentence: "In summary, our case report documents the medical phenomenon of persisting SARS-CoV-2 infection in an immunocompromised patient with impaired humoral and cellular immune response." Page 18, Line 560 – 1*

*Furthermore, we checked throughout the manuscript including abstract to point out the impaired function of presumably both the humoral and cellular immune response.*

*We agree that the patient presents a very low total IgG concentration, probably due to the B-cell depleting therapy with rituximab and may impair the evaluations of specific antibody testing by low total IgG counts. However, in the avidity assay, the spike-specific IgG concentrations were adjusted to 100 IU/ml to avoid bias by low antibody concentrations as described in methods and references.*

*However, in the cellular assay the PBMC counts were adjusted to 1 x $10^6$ cells/mL to avoid bias by different cell numbers. However, an impairment of cellular reactivity is likely caused by immunosuppressants and the underlying malignant condition of the patient and low CD4+ naïve T cell counts and CD4/CD8 ratio.*

5. Discussion: Line 506-507: The authors have written 'With the aid of NGS, we witnessed the directed mutational changes of SARS-CoV-2 probably enforced by exclusively humoral immune defence without destruction of infected cells by cell mediated immunity.' The authors may change this sentence as 'With the aid of NGS, we witnessed the directed mutational changes of SARS-CoV-2, probably enforced by insufficient humoral immune defence.

*Authors: Dear Reviewer, thank you for this improvement. We changed the sentence according to your specifications. Page 18, line 563*

Minor points:

Introduction

Line 44: EDCD? Please write before abbreviation (European Center for Disease prevention and control).

Authors: We fully agree and now added the correct name before abbreviation. Line 18, page 2

Line 57: convalescent plasma? What is the difference from convalescent plasma?

Authors: Thank you for this advice. We changed it to the correct term "convalescent plasma" throughout the manuscript.

Line 61-63: Two of the cases, patients treated....

Authors: We changed the beginning of the sentence to: "Two of the patients treated with monoclonal and convalescent plasma showed unusually high numbers of nucleotide changes and deletion mutations. Page 2, Line 36-7

2. Statistics: $p=0.05$  $p<0.05$,

Authors: Thank you, for pointing this out. We fully agree and changed it. Page 22, line 722.

3. Intratect  Please write IVIG by not indicating the brand.

Authors: We agree and changed it accordingly. Page 7, line 218 and 223

4. Discussion:

Line 269 (Results), 433, 459, 460: domaine  domain

Authors: Thank you for this suggestion. We corrected the term throughout the manuscript.

Line 485: What does nsp stands for?

Authors: nsp is the abbreviation for non-structural protein. The term is first mentioned on Page 18, line 519 and the abbreviation is explained there: "ORF1a and ORF1b are coding regions for non-structural proteins (nsp) (Rohaim et al., 2021)."

5. Serology:

Line 264: ELNA, CLIA  Please write before abbreviation.

Authors: We fully agree and added the full terms in lines 586 und 599, page 19.

Reviewer #2 (COVID-19, viral immunity) (Remarks to the Author):

This manuscript describes an interesting case of a single immunocompromised individual who experienced prolonged SARS-CoV-2 infection with shedding of infectious virus. The authors demonstrate time-dependent intra-host evolution of SARS-CoV-2 including the emergence of multiple mutations associated with immune escape that have been found in VOI/VOCs. Additionally, the emergence of escape mutations primarily occurs later in disease in the setting of an improved humoral immune response (based on serology testing) that is sufficient to decrease viral titres without eradication. The main conclusion is that prolonged viral replication in immunocompromised hosts can be a source of viral evolution, including the selection of specific mutations associated with immune escape.

The mechanisms driving SARS-CoV-2 evolution and the selection of immune escape mutations is of broad interest. However, there are now many reports of SARS-CoV-2 viral evolution in immunosuppressed hosts (For example - PMID 33176080, 33831372, 33915337, 34278371, 34737266, 34431691, 34319130, 33248470, 33545711). Many of these reports have documented the emergence of viral variants harboring mutations associated with VOI/VOCs and/or humoral immune escape,

including many mutations described in this manuscript. Given that this phenomenon has now been reported multiple times and many of these reports include detailed immunologic characterizations and have identified the emergence of escape mutations, this case adds support to what is already known rather than describing a new finding.

Authors: We agree that this phenomenon has been reported several times. We added the references as well as the paragraph: “

Meanwhile, there are more studies that shed light on the evolution of immune escape variants in immunocompromised patients and support the results of our study (12, 62-68). Nonetheless, our study not only shows the accumulation of an unusually high number of immune escape mutations in a single patient, which to a strikingly high degree evolved in parallel in various variants of concern. The chronology of mutation events during 7 months of infection shows a rapid accumulation of non-synonymous mutations which in part were persistent, in part temporary or even repeatedly acquired and lost.” Page 18, Line 549-53.

Other recommendations:

- As there are other reports of in-host evolution of escape mutations during infections in immunosuppressed individuals the title “first evidence” is incorrect

Authors: Thank you for giving our work your valuable time, attention and expertise.

We agree that in the meantime, to some extent also due to the long handling process, it is not a “first evidence” anymore. However, in view of the dramatic dynamic of the pandemic with occurrence of VOC, we reviewed our previously submitted manuscript in the light of the appearance of the omicron variant and found an astonishing similarity to this VOC. Therefore, we reanalysed our case and revised our manuscript in view of new VOCs and propose to change the title to “The mutational steps of SARS-CoV-2 to become like Omicron within seven months: the story of immune escape in an immunocompromised patient”. We have updated the manuscript to include the comparison with the Omicron variant throughout the manuscript.

We think that this comparison is indeed a novelty to describe the mutational steps in an individual patient. 50% of the non-synonymous mutations acquired by the investigated strain in our patient are also described for Omicron, and 88.2% of all acquired mutations have already been described in a VOC/VOI. In fact, we found that the chronology of immune escape in our individual patient independently evolved many steps that are now present of the overall SARS-CoV-2 population, to become like Omicron and highly similar to the sum of all variants of concern within 7 months.

- SARS-CoV-2 T cell studies should include additional individuals with or without previous SARS-CoV-2 infection as positive and negative controls to ensure that the SARS-CoV-2 peptide pools / assay is functioning as expected.

Authors: Dear Reviewer, we thank you for this important comment.

For the ELISpot assay a commercially available assay was used and performed according to the manufacturer’s protocol.

We clarified the assessment criteria and added a paragraph to describe our standardized laboratory procedures to avoid bias by inter-assay-variations. Pre-pandemic PBMC samples were used to determine the lower cut-off

The assessment criteria for the ELISpots were a minimum of 50 SFU in the positive control and a maximum of 10 SFU in the negative control according to the manufacturer's definitions [Pantaleo G, Harari A. Functional signatures in antiviral T-cell immunity for monitoring virus-associated diseases. Nat Rev Immunol. 2006 May;6(5):417-23. doi: 10.1038/nri1840. PMID: 16622477]. When those criteria were fulfilled, the stimulation index (SI) was calculated by dividing the mean SFU numbers in the antigen-specific wells with the mean SFU numbers of the negative control. The test was assessed negative with an SI < 2 according to previous determination of the cut-off by well-defined pre-pandemic PBMC samples and by PBMCs from SARS-CoV-2-naive individuals. The test was suggested to be poorly reactive with an SI between 2 and 7 and reactive with an SI ≥ 7 as defined by the manufacturer [Pantaleo G, Harari A. Functional signatures in antiviral T-cell immunity for monitoring virus-associated diseases. Nat Rev Immunol. 2006 May;6(5):417-23. doi: 10.1038/nri1840. PMID: 16622477] and samples of patients with acute infection. According to standardized laboratory procedures, in each assay, a standard laboratory control sample of a high-reactive and a non-reactive PBMC sample, respectively, was run to determine inter-assay-variations. Only assays with less than two standard deviations of the high-reactive and the non-reactive PBMC control sample, respectively, were defined valid. Page 19, line 609 ff.

- Editing could significantly improve the presentation and clarity of the manuscript

Authors: We have updated the presentation of the manuscript to include the Omicron variant and made several language and terminology improvements. Furthermore, we have been able to include all the reviewers' valuable suggestions for corrections, which, we hope, greatly improves the manuscript. The title has been changed and all figures have been updated.

REVIEWER COMMENTS

Reviewer #1 (Remarks to the Author):

The authors revised the manuscript according to the suggestions.

There are a few concerns regarding the manuscript.

I could not find the previous version of the manuscript, albeit the patient's clinical features were written in more detail.

In this revised version, I could not find the age (around 60?), gender (female?), fate (death?). These, if known should be added.

Therapeutic algorithms and guidelines were prepared in every country's Ministry of Health (etc.

<https://files.covid19treatmentguidelines.nih.gov/guidelines/covid19treatmentguidelines.pdf>)

Antiviral therapy is present in many of them. What's the situation in the country of the patient?

The vaccination state of the patient is also related with the humoral data and viral evolution. How many vaccinations does she have?

The authors answered one of the questions regarding antiviral therapy as:

We did not used... (page 7, line 225-26).

I could not find the related sentence in the referred place and not understand why the authors did not give antiviral therapy for a patient with lymphocytic lymphoma, severe COVID with persistent SARSCoV2 PCR positivity.

As convalescent plasma is used, it may also be added to Fig.4 and 5.

Fig.2 and 3. may be combined as both show similar data.

Page 3, line 53: METHODSRESULTS?

Page 3, line 68: with little white expectorate... nonpurulent secretion...

Page 3, Line 86: to date ???  for ... months.

Page 4, Line 100-103: Sentence to be corrected.

Line 102: CLIA?  Missing explanation before abbreviation.

Page 4, Line 104: Humoral immune responses increased in the course of disease...

How can the authors think about an increase when the patient was given IVIG and convalescent plasma?

Page 6, Line 172: Column 1 is present in both Table 1A and B.

Page 8, Line 225: The authors better write number instead of percentages.

Page 5, Line 147-50: Are there are 26, 22 substitutions, or 17.

Page 11, Line 309: Four ... serological methods..., namely ...,...

Page 11, Line 313-314: The sentence "Cellular immunity was diminished..." needs clarification here.

Page 12, Line 322-3: The decrease in 'evolutionary rate' could not be interpreted from Fig.5. New mutations were seen after day 124 in the figure.

The authors may better not use triple dot in tables and figures.

Reviewer #2 (Remarks to the Author):

In this revised manuscript the authors describe an interesting case of a single immunocompromised individual who experienced prolonged SARS-CoV-2 infection with shedding of infectious virus. The authors demonstrate a significant amount of time-dependent intra-host evolution of SARS-CoV-2 including the emergence of many mutations associated with immune escape that have been found in VOI/VOCs. The main conclusion is that prolonged viral replication in immunocompromised hosts can be a source of viral evolution, including the selection of specific mutations associated with immune escape.

The authors have addressed all of my concerns. However, there are numerous grammatical errors and I would suggest further editing to improve the presentation and clarity of the manuscript.

For example:

A few examples include:

- but the leukocyte typing showed a decline "by the" B-CLL of 70%
- The patient again received "the" IVIg therapy
- An increase of leukocytes "was developed" (21,600/ μ L) - this sentence abruptly ends a paragraph without explanation.
- Since the onset of "the" symptoms
- This certainly left us with the impression of "mutational escape manoeuvres". (what is a mutational escape maneuver?)
- We thus suggest that the accumulated mutations are "results" of an increased selection pressure on spike

REVIEWER COMMENTS

Reviewer #1 (Remarks to the Author):

The authors revised the manuscript according to the suggestions.
There are a few concerns regarding the manuscript.

I could not find the previous version of the manuscript, albeit the patient's clinical features were written in more detail. In this revised version, I could not find the age (around 60?), gender (female?), fate (death?). These, if known should be added.

Authors: We agree and added the following: sex and approximate age in line 87 as well as the sentences regarding fate "After seven months of continuous COVID-19 infection with a mild course of disease and confirmed pharyngeal PCR tests, the viral load became progressively lower and the viral infection was finally cleared. The patient is now considered to have recovered from COVID." (page 3; line (87-89). For reasons of anonymous presentation of the case the definite age cannot be given.

Therapeutic algorithms and guidelines were prepared in every country's Ministry of Health (etc. <https://files.covid19treatmentguidelines.nih.gov/guidelines/covid19treatmentguidelines.pdf>)

Antiviral therapy is present in many of them. What's the situation in the country of the patient?

Authors: We agree that the guidelines may be country-specific and, thus, should be stated in the manuscript. The guidelines for the treatment of mild courses in home quarantine can be found here: <https://www.awmf.org/leitlinien/detail/ll/053-054.html>, unfortunately in German. Antiviral treatment of COVID patients was reserved exclusively for inpatients and ICU patients in the Austrian guidelines. Here you can find the Austrian guideline for patients with severe courses or those in whom a severe course was expected: https://www.awmf.org/fileadmin/user_upload/Leitlinien/113_Internistische-Intensiv-Notfall/113-001LGk_S3_Empfehlungen-zur-stationaeren-Therapie-von-Patienten-mit-COVID-19_2022-03.pdf. These recommendations include sotrovimab, remdesivir, etc. However, the course of the patient we accompanied was consistently mild and the treating doctors decided against therapies with which they did not have much experience at that time.

The vaccination state of the patient is also related with the humoral data and viral evolution. How many vaccinations does she have?

Authors: In June 2021, after six months of persistent viral shedding, two doses of COVID-19-mRNA vaccines (BNT162b2; Comirnaty[®], BionTech/Pfizer) were administered. We added the information on page 3, line 80 - 82).

The authors answered one of the questions regarding antiviral therapy as:

We did not used... (page 7, line 225-26).

I could not find the related sentence in the referred place and not understand why the authors did not give antiviral therapy for a patient with lymphocytic lymphoma, severe COVID with persistent SARSCoV2 PCR positivity.

Authors: We had added the sentence: "No antiviral therapeutics were administered to the patient at any time of the infection. (page 3, line 78)." And apologize for the incorrect designation of the line. The treating physicians referred to the Austrian guidelines for COVID-19 treatment <https://www.awmf.org/leitlinien/detail/ll/053-054.html>, which recommend antiviral treatment only for severe cases, which our patient fortunately never had. We think that this fact, that the course of

disease was consistently relatively mild, is an interesting detail in the history of the strain mimicking Omicron and should be emphasized more strongly. Therefore, we have added the information about the mild course in the above mentioned sentence: (page 3; line 87 - 91)

As convalescent plasma is used, it may also be added to Fig.4 and 5.

Authors: The patient received RTX at two dates and IVIG at three dates. All these dates are now shown in Figures 1, 4 and 5.

Fig.2 and 3. may be combined as both show similar data.

Authors: We agree and combined figures 2 and 3.

Page 3, line 53: METHODSRESULTS?

Authors: We apologize and changed the term to RESULTS.

Page 3, line 68: with little white expectorate... nonpurulent secretion...

Authors: We agree and changed it.

Page 3, Line 86: to date ???  for ... months.

Authors: We agree and changed to entire paragraph into: "Persistent viral shedding was determined via qPCR at 25 time points in a 207 days lasting period of observation (Fig. 1). After seven months of continuous SARS-CoV-2 infection with a mild course of disease and confirmed pharyngeal PCR tests, the viral load became progressively lower and the viral infection was finally cleared. The patient is now considered to have recovered from SARS-CoV-2 infection. The patient gave full written consent for the case to be attended and published." Page 3, line 86 ff.

Page 4, Line 100-103: Sentence to be corrected.

Authors: The first serological tests were performed 102 days after symptom onset. At this time, slightly positive values of 17.7 AU/mL were measured in the chemoluminescent immune assay CLIA, as well as borderline neutralisation titres of 1:4 in enzyme linked neutralisation assay (ELNA). Page 4, line 107 – 110.

Line 102: CLIA?  Missing explanation before abbreviation.

Authors: We agree and thank you for the comment.

Page 4, Line 104: Humoral immune responses increased in the course of disease...

How can the authors think about an increase when the patient was given IVIG and convalescent plasma?

Authors: We thank the reviewer for raising this question and apologize that we did not clearly point out these aspects. The patient never received convalescent plasma and the IVIG batches were identified in consultation with the manufacturer as being free of specific antibodies against SARS-CoV-2 or SARS-CoV-1 due to pooling pre-pandemic plasma preparations. This information is added to the description of the patient course (see line 78ff page 3). Therefore, we interpret the increase of specific antibodies, measured via chemoluminescence assay, ELISA, immunoblot and neutralization test as patient-derived.

Page 6, Line 172: Column 1 is present in both Table 1A and B.

Authors: Thanks for this advice. In the table legend we subdivided into Columns 1 A and B ... page 6, line 172 ff.

Page 8, Line 225: The authors better write number instead of percentages.

Authors: We agree and changed the entire paragraph: "Overall, 17 of the 22 mutations (77.3%) acquired by the investigated strain convergently evolved in other VOC, mainly in the Alpha and Omicron variants. In the spike-coding region, the proportion of acquired mutations identical to mutations of VOC is even higher - 15 of the 17 mutations acquired there (88.2%) are found in other VOC. Overall, SARS-CoV-2 developed eleven persistent mutations during the study period of 140 days as well as eleven temporary mutational events. The chronology of intra-host mutational events is displayed in Fig. 3 and 4. An overview of the total number as well as a characterization of mutations accumulated by the investigated strain during the 7-month study period are shown in Table 2." Page 8, line 186 ff.

Page 5, Line 147-50: Are there are 26, 22 substitutions, or 17.

Authors: The sentences 147-150 seem to cause confusion. We have deleted one sentence and shortened the list of all acquired substitutions. A precise, correct and - in our opinion - clear listing of all acquired mutations of the strain during the observation period is depicted in Table 2:

Table 2: Listing of all persistent and temporary non-synonymous mutations that the strain has accumulated over the 7-months study period.

	n total	VOC/VOI	[%]	n in spike	VOC/VOI	%
Acquired non-synonymous mutations:	22	17	[77.3]	17	15	[88.2]
Persistent non-synonymous mutations:	11	8	[72.7]	9	8	[88.9]
temporary non-synonymous mutations:	11	9	[81.8]	8	7	[87.5]

n total, number of all non-synonymous intra-host acquired mutations; n in spike, number of all non-synonymous intra-host acquired mutations in the spike-coding region; VOC/VOI, number of mutations that are found in comparable expression in a variants of concern (VOC) or a variant of interest (VOI).

In total, the strain developed 22 mutations, 11 of it were persistent, 11 were temporary. 17 of the 22 acquired mutations occurred in the region coding for spike, 9 of those 17 spike – mutations were persistent, 7 were temporary.

Page 11, Line 309: Four ... serological methods..., namely ...,...

Authors: We added the listing of the serological methods: Four different well-established serological methods, namely CLIA SARS-CoV-2 TrimericS IgG, Microarray immunoblots, Neutralization test and IgG-anti-spike ELISA with the evaluation of the IgG-anti-spike-avidity gave insights into the humoral immune response and demonstrated the inability to clear the SARS-CoV-2 infection despite positive antibody responses. Page 11, line 311 – 313.

Page 11, Line 313-314: The sentence "Cellular immunity was diminished..." needs clarification here.

Authors: We added the sentence: "Indications for a reduced SARS-CoV-2-specific cellular immunity were given by the negative IFN- γ ELISpot. " Page 11, line 299 - 300.

Page 12, Line 322-3: The decrease in 'evolutionary rate' could not be interpreted from Fig.5. New mutations were seen after day 124 in the figure.

Authors: We agree. There was no measurable change in the evolutionary rate. Therefore, we changed the sentence the following way: "Interestingly, the evolutionary rate stayed constant throughout the first months of infection and no significant change was measured after the increase of specific

antibodies on day 124." Page 11, line 314 ff.

The authors may better not use triple dot in tables and figures.

Authors: We agree and have replaced triple dots by commas..

Reviewer #2 (Remarks to the Author):

In this revised manuscript the authors describe an interesting case of a single immunocompromised individual who experienced prolonged SARS-CoV-2 infection with shedding of infectious virus. The authors demonstrate a significant amount of time-dependent intra-host evolution of SARS-CoV-2 including the emergence of many mutations associated with immune escape that have been found in VOI/VOCs. The main conclusion is that prolonged viral replication in immunocompromised hosts can be a source of viral evolution, including the selection of specific mutations associated with immune escape.

The authors have addressed all of my concerns. However, there are numerous grammatical errors and I would suggest further editing to improve the presentation and clarity of the manuscript.

For example:

A few examples include:

- but the leukocyte typing showed a decline "by the" B-CLL of 70%
- The patient again received "the" IVIg therapy
- An increase of leukocytes "was developed" (21,600/ μ L) - this sentence abruptly ends a paragraph without explanation.
- Since the onset of "the" symptoms
- This certainly left us with the impression of "mutational escape manoeuvres". (what is a mutational escape maneuver?)
- We thus suggest that the accumulated mutations are "results" of an increased selection pressure on spike

Authors: Dear Reviewer, we sincerely thank you for your approving words and your suggestions for improvement. We have both made the grammatical changes you noted and consulted a native speaker from the academic field for review and correction. We sincerely hope that you appreciate our efforts and agree with the improvements.